# Rethinking Attention Mechanisms in Vision Transformers with Graph Structures

**DOI:** 10.3390/s24041111

**Published:** 2024-02-08

**Authors:** Hyeongjin Kim, Byoung Chul Ko

**Affiliations:** Department of Computer Engineering, Keimyung University, Daegu 42601, Republic of Korea; henryjoshuakim@gmail.com

**Keywords:** vision transformer, graph head attention, multi-head attention, graph attention network, lightweight model

## Abstract

In this paper, we propose a new type of vision transformer (ViT) based on graph head attention (GHA). Because the multi-head attention (MHA) of a pure ViT requires multiple parameters and tends to lose the locality of an image, we replaced MHA with GHA by applying a graph to the attention head of the transformer. Consequently, the proposed GHA maintains both the locality and globality of the input patches and guarantees the diversity of the attention. The proposed GHA-ViT commonly outperforms pure ViT-based models using small-sized CIFAR-10/100, MNIST, and MNIST-F datasets and a medium-sized ImageNet-1K dataset in scratch training. A Top-1 accuracy of 81.7% was achieved for ImageNet-1K using GHA-B, which is a base model with approximately 29 M parameters. In addition, with CIFAR-10/100, the existing ViT and parameters are reduced 17-fold and the performance increased by 0.4/4.3%, respectively. The proposed GHA-ViT shows promising results in terms of the number of parameters and operations and the level of accuracy in comparison with other state-of-the-art ViT-lightweight models.

## 1. Introduction

Transformers are among the most powerful neural network tools and have shown a promising performance using sequential data when applied to natural language processing (NLP) [1] and speech recognition [2]. A vision transformer (ViT) [3], applied in the field of computer vision, is a leading algorithm used in various vision problems, including image classification [3], image segmentation [4], object tracking [5], depth estimation [6], and action recognition [7]. ViTs are an important approach because they can be used for image processing without significantly changing the overall transformer architecture. Using a transformer with a sequence of image patches applied as an input, and without the need for a convolutional neural network (CNN), ViTs have achieved a performance exceeding that of CNN-based models. Despite such a high performance, a ViT treats all the tokens equally and ignores locality, thus losing the unique local structure of an image. In addition, ViTs still have a disadvantage in that it is difficult for them to process high-resolution images because of the large number of training data and self-attention operations. To address these problems, several studies have been conducted on improving the transformer architecture. Such studies can be divided into attempts at reducing the number of training data [8,9], operations, and parameters [10,11,12], as well as maintaining the locality of an image [13,14,15]. To reduce the number of training data, the DeiT model [8] applies a competitive convolution-free transformer with limited training data. This method relies on distillation tokens in introducing a teacher–student strategy specific to the transformer, allowing students to pay attention and learn from the teacher. Liu et al. [9] empirically analyzed various ViTs and showed that convolution with vision transformers is generally much more effective at generalization to fewer data. They also proposed a self-supervised assistant task to normalize the ViT training. A gated multi-layer perceptron (gMLP) [10] applies a simple network architecture to reduce the numbers of operations and parameters, performing as well as a transformer used in natural language and vision applications. A gMLP has experimentally achieved the same level of accuracy but without the self-attention of a ViT, thus demonstrating the lack of importance of self-attention. By replacing the self-attention sublayer with a simple linear transform that mixes with the input token, FNet [11] can accelerate the transformer encoder architecture with a limited decrease in accuracy. In particular, the training time is significantly reduced by replacing the self-attention with a Fourier transform without learnable parameters. To solve the problem of high-resolution image processing, owing to the large number of self-attention operations, PVT [12] reduces the number of operations of large feature maps using progressively size-reduced pyramids. To maintain the locality of the image, a Swin transformer [13] applies a shifted window consisting of several patches within each window, and the self-attention is calculated solely for this window. This enhances the locality of the transformer because the attention is calculated only for the region bounded by the window and not for the entire area. CeiT [15] applies an image-to-token module to extract patches from the low-level features generated instead of simple tokenization from raw input images. LocalViT [16] not only applies a depth-wise convolution to the feedforward network to reduce the computational load but also incorporates a wide range of design choices, such as an activation function, a layer placement, and an expansion ratio, as a way to provide locality mechanisms.

However, such ViT-based methods do not properly consider the spatial geometric relationship between local regions or between global and local regions and have limitations in reducing the number of computations because they depend heavily on the combination of multi-head attention (MHA). In addition, because ViTs require a large number of training data, their performance is significantly reduced when trained using small datasets without a pretraining step.

### Contribution of This Work

In this study, to reduce the number of ViT operations and preserve the global and local features of the image classification, we introduce a new graph head attention (GHA) mechanism for ViT that replaces MHA with fewer graph heads using the proposed graph generation and graph attention. From this, we show that GHA works effectively on small datasets and outperforms SoTA models on conventional datasets. Our contributions are as follows:

Unlike with other graph-based transformers [17,18,19], which apply graphs and attention in parallel and combine the outputs, this study is the first attempt to apply a graph inside the transformer head and replace MHA with a few GHA mechanisms. Moreover, there is no need for a class token in patch embedding, and thus the number of operations can be reduced.The links of nodes with low attention scores are excluded using graph pooling, and the node features are updated by applying GHA boosting to reflect the connectivity of the neighboring nodes. This process preserves the feature locality and secures the diversity of the attention. GHA-ViT not only creates tokens with local characteristics but also learns the relationship between tokens using a graph structure, which eventually strengthens the locality of the tokens.Because the graph structure is constructed using an attention matrix and the node feature are extracted from a value matrix, additional learning parameters for a graph construction are not required.GHA-ViT shows a promising classification performance with only scratch training conducted on small and medium-sized datasets and no pre-training on large datasets.

Figure 1 shows the overall architecture of the proposed GHA-ViT model.

## 2. Related Studies

Many studies have applied graphs to transformers [20,21,22]. A graph transformer network (GTN) [20] was proposed to exclude noisy connections and include useful connections to tasks (e.g., metapaths) while learning effective node representations from the new graphs in an end-to-end manner to create new graph structures. Graphormer [21] utilizes a graph as a transformer to effectively encode the structural graph information into the model. With this model, centrality encoding, which can capture the node importance of the graph, and spatial encoding, which can capture the structural relationship between the nodes, are applied inside the transformer encoder. Yun et al. [22] proposed a GTN with another structure, which excludes noisy connections between graph nodes while preserving the effective nodes within the transformer. A graph-oriented transform (GraFormer) [23] was proposed for 3D pose estimation. GraFormer not only fuses the information on the graph nodes by repeatedly stacking graph attention blocks and graph convolution layer blocks but can also model the topological structure of the graphs. However, the aforementioned graph transformer models are inapplicable to vision applications because they are designed for node classification (e.g., citations, movies, traffic, social networking, and protein interactions) and human poses rather than for images.

### Graph Vision Transformer Models

Unlike graph transformers being applied to general node classification, relatively few studies have applied graphs to ViTs for vision applications. Shen et al. [17] proposed a graph interactive transformer (GiT) for vehicle reidentification. Using this method, the GiT is divided into two modules: the original transformer module for extracting powerful global patch features and a local correlation graph (LCG) module for extracting local features that are distinct within the patch. The output features of the LCG and transformer modules are combined and used for downstream tasks. Zheng et al. [18] proposed a graph transformer network, which is a graph representation of an entire slide image, as well as a method for fusing transformers. The image patches become a set of nodes in the graph, and each node is connected by an edge. The constructed graph is put into the transformer using a graph convolutional network (GCN) and a pooling layer. The output of the transformer is applied to the MLP head for classification. Mesh Graphormer [19] integrated graph convolution with self-attention as a way to reconstruct human poses and meshes from a single image. It was proven that both the graph convolution and grid features of Mesh Graphormer helped improve the performance of a pure transformer. The vision graph neural network (ViG) [24] was the first to combine graph structures with images. It regards each image patch as a single graph node and employs k-NN to build the relations between each image patch. Despite the novelty of this approach, it has the disadvantage of capturing only the similarities of the image patches without considering the latent image structure. Graph and optimization-based heterogeneous structured pruning (GOHSP) [25] applied graphs to optimize the model structure. However, because the interest was only in making the model structure more lightweight, the learning limitations of the model, such as the inductive bias, have not been improved.

In both research categories, a graph is not directly applied to the attention, and the existing attention head and graph are combined. In this case, frequent feature dimension scaling occurs for the feature combination, and the output is not a true combination of the global and local features. In addition, because MHA is used as is and a graph is added as a separate module, the number of operations can be increased. The remainder of this paper is organized as follows. Section 3 provides a brief description of the ViT and graph neural networks. We then present the details of the proposed GHA-ViT in terms of the architecture, training, and testing in Section 4. Section 5 describes the experimental environment and provides a comprehensive evaluation of the proposed method based on the results of various experiments. Finally, some concluding remarks are given in Section 6.

## 3. Preliminaries

### 3.1. Vision Transformers

Inspired by the success of the transformer scaling used in NLP, the ViT [3] directly applies a standard transformer to an image with minimal modifications. To achieve this, the ViT splits the image into P patches (such as a token in NLP) with a feature dimension of d and feeds a sequence of linear embeddings of these patches as input to the transformer. After the embedded patch matrix $H\;\in R^{P\times d}$ is normalized, it passes through the MHA module and is fed to the MLP to generate the output of the encoder. The encoder consists of L layers and is iteratively operated from 1 to L. Finally, the representation y is obtained using calculations using the prediction head weight $W_{c}\;\in \;R^{d\times \left|C\right|}$ and bias $b_{c}\;\in \;R^{\left|C\right|}$, where |C| is the number of classes.

### 3.2. Patch Generation

Pure ViTs apply a single convolutional-layer-based image tokenizer to place a given image into a small patch. They divide an image into several patches according to the given patch size without a replacement. In this regard, each patch has fundamental difficulty in securing the locality of the image. It is therefore necessary to learn a significant number of datasets to understand the inductive bias. In this study, we employed a pooling-layer-based tokenizer [26] to overcome this limitation. The pooling layer, which is helpful in enhancing the inductive bias, aids in escaping the big data paradigm.

### 3.3. Graph Attention Networks

A graph attention network [27] is a convolution-style neural network that can be applied to graph-structured data using a masked self-attentional layer. Unlike a GCN [28,29], a GAT assigns different importance to nodes in the same neighboring group and can therefore improve the model capacity and help with the interpretation. This structure enables inductive learning because such learning is possible without access to the entire graph. The attention score redefines the input data by determining the importance of the neighbor of node i as follows:(1)$$h_{i}^{'}=\sigma (\alpha_{ii}h_{i}W+\sum_{j\in N\left(i\right)}\alpha_{ij}h_{j}W)$$
where N(i) is the number of nodes neighboring node i, α is the normalized attention score, and $W\in R^{d\times d^{'}}$ is the trainable parameter for the node feature $h\in R^{n\times d}$ (n is the number of nodes). σ is the activation function.

## 4. A Graph-Head-Attention-Based ViT

### 4.1. Graph Head Attention

The attention head is the core idea of a transformer and calculates the attention between each patch using three matrices: the query (Q), key (K), and value (V). The attention score is calculated as the scaled dot product of Q and the transposed K, and it can be executed in parallel. It therefore has an advantage over previous neural networks in terms of its computational speed. The transformer uses MHA; thus, n heads can learn different types of attention from the input and obtain a strong attention representation by combining them [3]. In reality, however, not all the heads of the MHA have the same effect on the attention performance of the transformer. Instead, only a part of the head affects the performance, and the remainder focuses on unnecessary parts that negatively affect the outcome of the final attention [30]. From this perspective, it is clear that MHA is inessential for a transformer. In this section, we propose a new transformer that can receive more attention with fewer GHA mechanisms and without the use of multiple heads.

Let X be the input of the encoder layer. Here, X consists of P patches, and the hidden dimension of the input patch is $d_{h}$. The i-th patch can then be denoted as $x_{i}\;\in \;R^{d_{h}}$ and $X=[x_{1},x_{2},\cdot \cdot \cdot ,x_{p}],X\;\in \;R^{P\times d_{h}}$. The Q, K, and V matrices have corresponding weight matrices $W_{Q},W_{K}$, and $W_{V}\in \;R^{d\times d_{h}}$, and Q, K and V can be obtained from the dot product of input X and the weight matrices. The attention matrix (AT) of the head is calculated as follows:(2)$$Q=XW_{Q}\;K=XW_{K},\;\;\;Q,K,V\;\in R^{P\times d_{h}}\;V=XW_{V}$$
(3)$$AT=softmax\left(\frac{QK^{T}}{\sqrt{d_{h}}}\right)\in R^{P\times P}$$

### 4.2. Graph Structure Generation

To provide better generalization and performance in a CNN, the pooling layer plays an important role in reducing the feature map size and broadening the receptive field. However, this pooling operation cannot be applied directly to a graph because there is no local information between the graph nodes. Therefore, inspired by the approaches described in [31,32], we propose graph pooling containing local information based on a mask filter. Graph pooling allows us to downsample the graph data and adaptively selects a subset of nodes to form new, smaller graphs. In this respect, we apply the *Top* − *k* function to the attention matrix AT to select sub-nodes with significant connectivity. Sparse distilled nodes can be regarded a new form of graph structure derived from an attention matrix. In other words, we consider the sparse matrix of these nodes to be the adjacency matrix AD of a graph. In the case of a structure with a total number of patches P and a total number of layers L, the value of k selected for each layer is increased by (0.8×P)/L from a minimum of 10% to a maximum of 80% of the total number of patches.
(4)AD=Top−k(AT,k)

We can now construct a graph consisting of node patches using AD. To consider the self-edge of the node, AD adds an identity matrix I. However, as shown in Figure 2a, directed and undirected edges are mixed into the initial graph constructed from AD. In a transformer, because the attention is created according to interactions with neighboring patches, a directed edge cannot guarantee the correct patch attention. Therefore, the mixed graph must be converted into an undirected graph. For this purpose, the following graph transformation method is proposed: First, as indicated in Equation (5), the upper matrix of AD and its transpose are added to form a partially undirected graph, as shown in Figure 2b:(5)$$AD_{triU}=triU(AD)+triU{\left(AD\right)}^{T}$$

Similarly, the lower matrix of AD and its transpose are added to form a partially undirected graph, as shown in Figure 2c:(6)$$AD_{triL}=triL(AD)+triL{\left(AD\right)}^{T}$$

Finally, the upper matrix $AD_{triU}$ and the lower matrix $AD_{triL}$ generate an undirected graph $\bar{AD}$ according to an OR operation ∨, as shown in Figure 2d.
(7)$$\bar{AD\;}=AD_{triU}\;\vee \;AD_{triL}$$

In addition, we employ a graph calibration process to render the graph robust to irregular connectivity. The graph intensity $AD_{I}$ and graph weight $AD_{w}$ are applied as follows:(8)$$\bar{AD\;}:\;=\left(\bar{AD\;}+\bar{AD_{I}}\right)⊙\;{\bar{AD}}_{w}$$
where $AD_{I},AD_{w}\;\in R^{P\times P}$ is a trainable parameter and ⊙ indicates a Hadamard product.

### 4.3. Graph Head Attention Boosting

To improve the accuracy of the image classification, detailed attention can be obtained using a MHA combination. However, MHA requires the weight matrices $W_{Q}$, $W_{K}$, and $W_{V}$ for each head. Therefore, as the number of heads increases, more learning parameters and a greater memory and computational time are required. In addition, because the attention of each head becomes similar when the number of heads exceeds a certain criterion, removing several attention heads during the test does not significantly affect the performance [30]. To avoid the problems caused by MHA, we used fewer and demonstrated that the transformer can operate successfully using only the proposed GHA. To ensure the diversity of attention, similar to MHA with fewer attention heads, and to emphasize the correlation between the graph nodes, we apply a GAT [27] to GHA. When using GHA instead of a general MHA mechanism, securing the diversity of the attention is an extremely important part of the successful operation of the GHA model. We previously extracted an $\bar{AD}$ representing the relationship between the node patches from the attention matrix. Herein, we apply the GAT to $\bar{AD}$ to achieve efficient attention computations between the nodes with $\bar{AD}$. From $\bar{AD}$ and V, the attention coefficient e between the nodes i and j is obtained using the learnable weight matrix $W_{c}$.
(9)$$e_{ij}=FN(W_{c}\;\cdot \;v_{i},W_{c}\;\cdot \;v_{j})$$

Here, an FN is a simple single-layer feedforward neural network that transforms the input into $R^{1\times d_{h}}\times R^{d_{h}\times 1}\;\to R$. The above expression indicates the importance of the features of the nodes i-j. At this time, j does not indicate all the nodes, only the neighbors N(i) of node i. Finally, if it passes the softmax function, the following normalized attention matrix $\tilde{AD}$ can be calculated:(10)$${\tilde{AD}}_{ij}=\frac{exp(e_{ij})}{\sum_{k\in N(i)}exp(e_{ik})}$$

By applying Equation (10) to all the encoder layers, GHA can dynamically consider the connections between the nodes to ensure the diversity of the node features. The final GHA is produced by applying the node feature matrix V and the weight matrix $W_{gat}$ of the GAT to $\tilde{AD}$ as follows:(11)$$\mathrm{GHA}(V)=\sigma_{2}(\tilde{AD\;}\cdot \;V\;\cdot \;W_{gat})$$
where $\sigma_{2}$ is a ReLU activation function. The value of GHA is obtained through the process of Figure 1c. Because the GHA encoder consists of L layers, V is put into the first layer, but from the second layer, V is changed to $h_{l}$, the output of each layer. GHA($h_{l-1}$) of the previous layer is again skip-connected (element-wise sum) with input $h_{l-1}$.
(12)$$h_{l}^{'}=GHA(h_{l-1})+h_{l-1},l\;\in \;\{1\cdot \cdot \cdot L\}$$

After $h_{l}^{'}$ is linearly normalized (LN) again and applied to the FFN, it is similarly skip-connected to the original $h_{l}^{'}$ to produce the output of the final encoder block, as shown in Figure 1a.
(13)$$h_{l}=\mathrm{FFN}\left(h_{l}^{'}\right)+h_{l}^{'},\;l\;\in \;\{1\cdot \cdot \cdot L\}$$

Finally, the output of the last encoder layer L, $h_{L}\;\in R^{P\times d_{h}}$, is passed to the readout layer. For the readout layer sequence pooling (seq) [26], the mean and max values were used to consider the diversity [33]. The multi-readout feature $H_{out}$ is calculated:(14)$$H_{out}=h_{seq}\;∥\;h_{mean}\;∥\;h_{max}$$
where ∥ denotes the concatenation operation between the node features. The multi-readout feature $H_{out}$, which has passed through the readout, is then classified using an MLP. The loss function was optimized using soft distillation [34,35].

## 5. The Dataset and Experimental Results

To evaluate the representation learning ability of the proposed GHA-ViT model, we compared it with VGG-16 [36], ResNet [37] and MobileNetV2 [38], which are representative CNN models; ViT-based methods [3,8,26,39]; MLP-based approaches [10,40,41]; and a graph-based method [24,25]. We proved through our experiments that the performance of the proposed model is similar to that of other state-of-the-art (SoTA) methods on several benchmark datasets. In addition, we demonstrated the effect of the graph-generating module through ablation experiments and demonstrated that a promising performance can be achieved even if fewer heads are used through attention visualization.

### 5.1. The Experimental Setup

**Datasets.** Various benchmark classification datasets are used to measure the capacity of the model. CIFAR-10/100 [42], MNIST [43], and MNIST-Fashion [44] are used as small-sized datasets, and ImageNet-1K [45] is used as a medium-sized dataset. Auto-Augment [46], Rand-Augment [47], and random erasing [48] were used as data augmentation methods.

**Baseline.** We set the baseline of the GHA differently to prove that the proposed GHA-ViT model can reduce the number of head and encoder layers. The basic structure of the GHA-ViT is based on DeiT [8] because it is inherently capable of learning with a small dataset. The baseline models are of two types, GHA-Base and GHA-Small, according to the number of heads and layers. Table 1 summarizes the GHA model used as a baseline. In addition to GHA-ViT, we used ResNet [37] as the CNN baseline model for the comparative experiments. The ResNet model has a modified last MLP layer to suit the number of classes for each experimental dataset. A pure ViT and DeiT are used as the transformer models for comparison with the GHA-ViT. These methods also have changed last MLP layers to obtain suitable outputs for each set of experimental data.

### 5.2. The Experiment Environment and Parameter Settings

We built a GHA-ViT model using the PyTorch framework. NVIDIA 3090Ti was used for the training. The following parameters were used in the model training. AdamW was used as the optimizer, and β1 = 0.9 and β1 = 0.999 were set as the values. The batch size was set to 128 for all training and testing. The initial learning rate was set to 0.005, and cosine warm-up and decay were used. The epoch was set to 300 for training, the input resolution of ImageNet was 224 × 224, and a 32 × 32 resolution was used for the small-sized datasets.

### 5.3. Comparing the Performance with State-of-the-Arts Models

Table 2 presents the performance comparison results of the GHA-ViT model for the small datasets when applying scratch training SoTA methods. GHA-ViT outperformed the SoTA methods for all types of small datasets. In addition, the GHA-ViT demonstrated efficient operation in terms of the learning parameters and operations. GHA-S, which is a small version of the proposed GHA-ViT, outperformed the SoTA methods on CIFAR-10 and CIFAR-100. GHA-B showed a 0.1% lower value than ResNet-18 on the MNIST dataset and a 0.1% lower value than ViT12/4 on the MNIST-F dataset but achieved the best performance for the CIFAR-10 and CIFAR-100 datasets. Owing to a low inductive bias, transformer-based models [3,26] achieve an overall low performance in an environment with small datasets. Compared with ViG-Ti [24], the proposed GHA-B also shows a higher accuracy of 3.8% for CIFAR-10, 6.0% for CIFAR-100, 0.3% for MNIST, and 0.3% for MNIST-F. These results confirm that the proposed GHA-ViT models perform well on small-sized datasets by maintaining the high locality and globality of the images when using the proposed graph structure. In terms of the number of parameters and operations, GHA-S uses approximately 17-fold fewer parameters than pure ViT-12/4 and approximately 5.8-fold fewer operations but shows a 4.3% higher accuracy on the CIFAR-100 dataset. GOHSP [25] demonstrates a 0.6% higher performance than the proposed GHA-B. However, unlike GHA-B, which applies only scratch training, GOHSP uses a pre-trained ViT model; therefore, a difference of 0.6% indicates that GHA-B can achieve a good performance without pre-training. In addition, GOHSP requires 5 M more parameters and 1070 additional units of computational complexity than GHA-S. In comparison with SAL-ViT [49], the proposed GHA also achieved a higher performance in the trade-off between computational volume and performance. This shows that the proposed GHA uses graphs to preserve the locality information more strongly and consume less computational costs than the attention approximation method. From the overall results, we can see that the proposed GHA-ViT method guarantees an inductive bias as high as that of a conventional CNN method and that it operates efficiently, even with a small number of training data.

We conducted experiments using the ImageNet-1K dataset to demonstrate that the proposed model works effectively on small- and medium-sized datasets. Table 3 presents the results of a performance comparison between the proposed GHA-ViT and other SoTA methods. Compared with ResNet-152, among the CNN-based methods [37], the number of parameters in the GHA-S model is reduced up to six times over, and the number of operations is reduced up to 6.4 times over. In terms of accuracy, the GHAS model is slightly inferior to ResNet-152, whereas the GHA-B model improves the Top-1 accuracy by 3.4%. In a comparison with ViT-based methods [3,8,14,26,39], in terms of accuracy, the GHA-S model has an increased Top-1 accuracy compared to that of PoolFormer-S12 under the same conditions (Param and MACs). In the case of GHA-B, the Top-1 accuracy was the highest at 81.7%; however, the number of operations was 1.1-fold higher than that of T2T-ViT-14 with a similar accuracy. Compared with MLP-based models [10,40,41], both the GHA-S and GHA-B models increased their Top-1 accuracy by approximately 2% on gMLP-Ti and gMLP-S, which have similar numbers of parameters and operations. In addition, in comparison with ViG methods [24] using graph structures, the numbers of parameters and operations are slightly higher, whereas the GHA-S and GHA-B models show high accuracies of 3.8% and 1.3%, respectively. This is because the proposed GHA-ViT model can generate a graph structure with a higher efficiency than the graph generation method used in ViG. When the GHA-B models are compared with GOHSP [25], the GHA models show a 1.8% higher Top-1 accuracy. In the case of the GHAS model, the Top-1 accuracy was 2.5% lower than that of GOHSP; however, the parameters and MACs were improved by 1 M and 1 G in comparison with GOHSP, respectively. In terms of the parameters and computational complexity, the proposed model is sufficiently competitive with other CNN and ViT-based approaches.

### 5.4. Ablation Studies

**Graph Structure Generation.** To verify the efficiency of the proposed graph generation method, we observed the difference in accuracy according to the change in the graph structure on various datasets, as shown in Table 4. Here, $G_{w/o}$ creates a graph using only the Top − k method from the attention map, and $G_{triU}$ creates a graph using an upper triangular matrix. In addition, $G_{triL}$ creates a graph using a lower triangular matrix. When $G_{all}$ generated graphs considering both the upper and lower triangular matrices, we obtained a higher accuracy than in the other three cases for all the datasets. As an interesting aspect of the experiment, the performance when only $G_{w/o}$ was used was higher than when only $G_{triU}$ and $G_{triL}$ were applied. This is because a complex graph structure maintains the association between the tokens without artificially breaking them. From the experiments, we observed that if we create a graph structure by considering only specific regions of the attention map, the graph cannot accurately reflect the overall characteristics of the image.

**Effect of the Multi-Readout Feature.** To prove the effectiveness of the multi-readout feature, we conducted a comparison experiment with various readout methods, as shown in Table 5. Table 5 indicates that the proposed multi-readout feature exhibits the highest performance for all the small datasets. From the results, we can see that the combined multi-readout features help classify the final image more effectively than using only the readout features obtained using simple operations, such as max or average operations.

**Type of Graph Convolution.** Table 6 presents the results when a representative graph convolution was applied to GHA-ViT. The GCN [28], GIN [50], and GAT [27] were used as graph convolution methods. As shown in Table 6, the proposed GHA-ViT method does not exhibit a large difference in the various graph convolution operations. This indicates that the GHA inherently has a highly flexible structure. Among the three graph convolutions, the highest accuracy was obtained when a GAT was used because a GAT assigns a different importance to patches in the same neighbor group, unlike a GCN or GIN, which treat all the relationships between patches in the same manner. In this study, a GAT was used as the default graph convolution when considering the experiment results.

### 5.5. Graph Visualization

To simplify describing the suitability of the proposed graph structure generation method, we visualized the graph structure generated from the GHA-ViT based on a small (S) model. Figure 3 shows the adjacency matrix created by the GHA-S and the graph generation method of a ViG [24], which generates an adjacency matrix by directly applying k-NN to the input image patches. As shown in Figure 3, the graph structure created using k-NN includes background patches that are unrelated to the target patch. When using the proposed graph generation method, only patches that have a strong relationship with the target patch were selected. This indicates that a graph generation method using an attention map is more effective than an image-based k-NN method. In addition, when we check the tokens generated by the proposed method, we can see that the tokens around the target token are selected. This indicates that the proposed graph generation method can preserve the feature locality.

## 6. Conclusions

In this paper, we proposed a new GHA method that can overcome the limitations of MHA, the core module of ViTs. By converting the attention map operation from a matrix perspective into a graph perspective, it is possible to significantly reduce the number of unnecessary operations and parameters while maintaining the accuracy of the image classification. We also demonstrated that the attention feature space embedded into multiple heads differed insignificantly from that when fewer graph heads were used. Through scratch training and experiments using various small datasets, the proposed GHA-ViT demonstrates promising results without being significantly affected by the number of datasets. In future studies, to improve the mask filter when constructing a graph and achieve a more meaningful attention output, we plan to apply a method combining different graph-pooling approaches such as graph U-NET [31]. Through such additional studies, the ViT performance of the GHA structure is expected to be significantly improved in comparison with that of MHA-based ViT approaches.

## Figures and Tables

**Figure 1 sensors-24-01111-f001:**
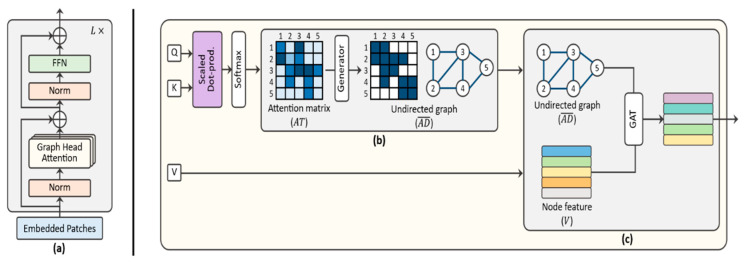
Overall architecture of the proposed GHA-ViT model: (**a**) GHA encoder layer composed of a few graph heads. (**b**) The attention score is calculated as the scaled dot product of Q and transposed K. Then, the graph generator is applied to the attention matrix AT for selecting sub-nodes and converted into an undirected graph $\bar{AD}$. (**c**) After graph generation, the value V and an undirected graph $\bar{AD}$ are applied to a graph attention network (GAT). Based on the generated graph and node features V, the GAT gives different importance to the neighboring nodes.

**Figure 2 sensors-24-01111-f002:**
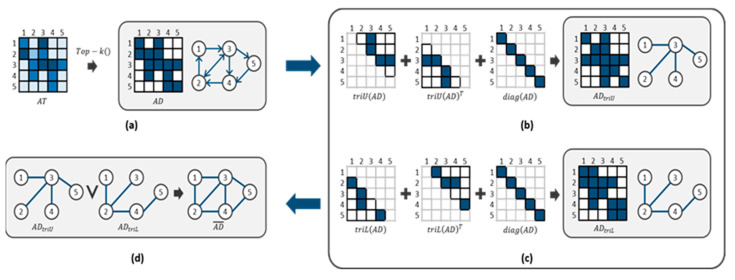
Graph generation process from attention matrix AT: (**a**) initial adjacency matrix and graph with threshold applied, (**b**) upper matrix $AD_{triU}$ and its graph, (**c**) lower matrix $AD_{triL}$ and its graph, and (**d**) undirected graph $\bar{AD}$. ∨ indicates OR operation.

**Figure 3 sensors-24-01111-f003:**
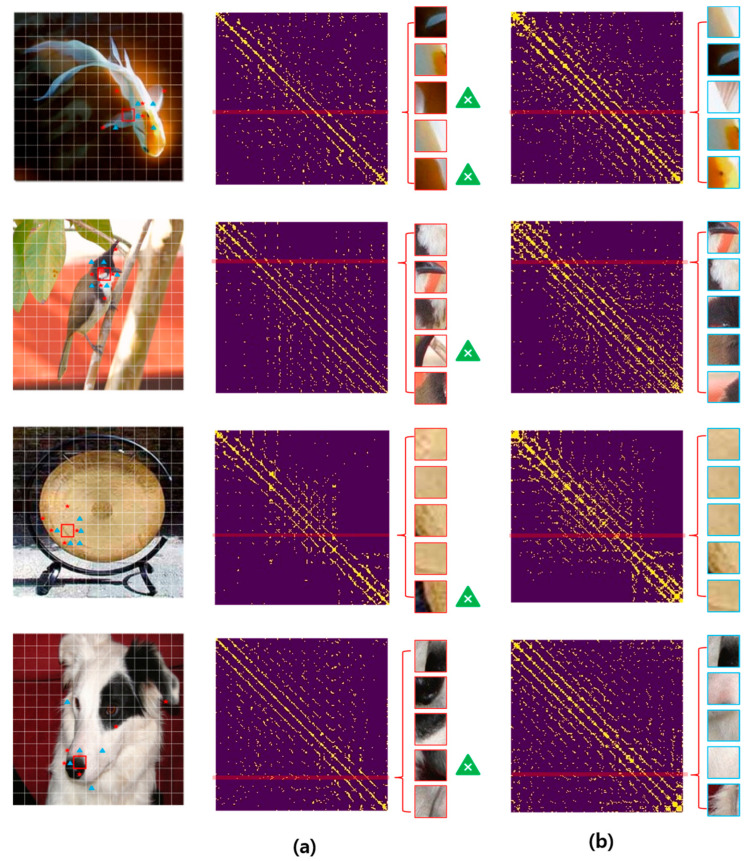
Graph structure visualization generated using the proposed graph generation method: (**a**) adjacency matrix generated using k-NN of the ViG [24] and (**b**) adjacency matrix using the GHA-S. The red line on the adjacency matrix indicates the row corresponding to the target patch (red star and box area) of the image. Patches on the right side of the adjacency matrix are patches that have the highest relationship with the target patch (yellow triangle and box area). As can be seen in Figure 3, the proposed method selects only patches that have a high relationship with the target patch, whereas the adjacency matrix generated using the k-NN method includes background images that have a low relationship with the target patch.

**Table 1 sensors-24-01111-t001:** Details on GHA-ViT model variants. “dim *d*” means hidden dimensions in encoders, and “mlp Ratio” means a scaling factor for hidden dimensions of MLP. In GHA-*-L/β, L means number of layers and β is patch size. * means model size (S: small, B: baseline).

Model	Patch Size (β)	Head	Layers (L)	dim *d*	mlp Ratio
GHA-S-7/3	3 × 3	3	7	64	2
GHA-B-7/3	3 × 3	6	7	64	2
GHA-S-14/7	7 × 7	3	14	64	4
GHA-B-14/7	7 × 7	6	14	64	4

**Table 2 sensors-24-01111-t002:** Performance comparison of scratch-trained CNN and transformer-based models on small-sized datasets. Image resolution is the same at 32 × 32.

Model	Params (M) ↓	MACs (M) ↓	Top-1 (%) ↑			
			CIFAR-10	CIFAR-100	MNIST	MNIST-F
VGG-16 [36]	20	155	90.1	70.7	99.7	94.6
ResNet-18 [37]	11	40	90.2	66.4	99.8	94.7
ResNet-34 [37]	21	80	90.5	66.8	99.7	94.7
ResNet-56 [37]	24	130	93.9	71.5	99.7	94.8
ResNet-110 [37]	43	260	94.1	72.6	99.7	95.1
MobileNetV2/0.5 [38]	1	10	84.7	56.3	99.7	93.9
MobileNetV2/2.0 [38]	8	20	91.0	67.4	99.7	95.2
ViT-12/4 ^†^ [3]	85	5520	94.8	74.1	99.6	95.4
ViT-Lite-7/16 ^††^ [26]	3	20	78.4	52.8	99.6	93.2
ViT-Lite-7/8 ^††^ [26]	3	60	89.1	67.2	99.6	94.4
ViT-Lite-7/4 ^††^ [26]	3	260	93.5	73.9	99.7	95.1
CVT-7/8 [26]	3	60	89.7	70.1	99.7	94.5
CVT-7/4 [26]	3	250	94.0	76.4	99.7	95.3
CVT-7/3 × 2 [26]	3	1290	95.0	77.7	99.7	95.1
ViG-Ti [24]	6	1230	93.0	74.1	99.4	95.0
GOHSP ^†††^ [25]	10	2020	97.4	-	-	-
SAL-ViT [49]	5	1601	95.9	77.6	-	-
**GHA-S-7/3**	5	950	95.2	78.4	99.5	95.2
**GHA-B-7/3**	10	2130	96.8	80.1	99.7	95.3

^†^: Pure ViT consists of 12 multi-heads and 16 kernel sizes. However, because the image size is 32 × 32, the size of the kernel is reduced (to 4) in the same way as using other methods. ^††^: The ViT-Lite version used a different patch size on a 32 × 32 size input image. Experimental results of the ViT-Lite version are referenced from [26]. ^†††^: The GOHSP used pre-trained weights from training on ImageNet.

**Table 3 sensors-24-01111-t003:** Performance comparison of scratch-trained CNN and transformer-based models on the ImageNet-1K dataset. Image resolution is the same at 224 × 224.

Model	Params (M) ↓	MACs (G) ↓	Top-1 (%) ↑	Top-5 (%) ↑
ResNet-50 [37]	26	4.3	76.2	95.0
ResNet-101 [37]	45	7.9	77.4	95.4
ResNet-152 [37]	60	11.6	78.3	95.9
ViT-S-16 [3]	47	10.1	78.1	-
DeiT-S [8]	22	4.6	79.8	95.0
CCT-14/7 × 2 [26]	22	18.6	80.6	-
T2T-ViT-14 [14]	22	4.8	81.5	-
PoolFormer-S12 [39]	12	1.8	77.2	-
PoolFormer-S24 [39]	30	3.0	80.3	-
Mixer-B /16 [40]	59	12.7	76.4	-
ResMLP-12 [41]	15	3.0	76.6	-
gMLP-Ti [10]	6	1.4	72.3	-
gMLP-S [10]	20	4.5	79.6	-
ViG-Ti [24]	7	1.3	73.9	92.0
ViG-S [24]	23	4.5	80.4	95.2
GOHSP [25]	11	2.8	79.9	-
GHA-S-14/7	10	1.8	77.4	93.5
GHA-B-14/7	29	5.9	81.7	95.8

**Table 4 sensors-24-01111-t004:** Comparison of accuracy (%) according to the methods of graph structure generation.

Dataset	$G_{w/o}$	$G_{triU}$	$G_{triL}$	$G_{all}$
CIFAR-10	93.2	93.1	92.8	95.2
CIFAR-100	73.8	73.7	73.0	78.4
MNIST	99.4	99.3	99.4	99.5
MNIST-F	94.8	94.7	94.8	95.2

**Table 5 sensors-24-01111-t005:** Effect of the multi-readout feature. Comparison of accuracy (%) of different readout layers on small-sized datasets.

Dataset	Mean	Max	Seq	Multi
CIFAR-10	93.9	92.5	91.1	95.2
CIFAR-100	73.8	72.1	75.1	78.4
MNIST	99.2	99.4	99.2	99.5
MNIST-F	94.8	93.4	94.9	95.2

**Table 6 sensors-24-01111-t006:** Accuracy (%) difference for different types of graph convolution. The results from GHA-S on small-sized datasets.

Dataset	GCN	GIN	GAT
CIFAR-10	94.5	94.9	95.2
CIFAR-100	76.7	77.1	78.4
MNIST	99.4	99.4	99.5
MNIST-F	94.8	94.9	95.2

## Data Availability

Data are contained within the article.

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
