# Peer review of "Rethinking Attention Mechanisms in Vision Transformers with Graph Structures"

_sensors, 2024, doi:10.3390/s24041111_

Round 1

Reviewer 1 Report

Comments and Suggestions for Authors

1.The abstract needs to be further condensed, please highlight the innovation and novelty of the paper.

2. The algorithm selected for comparison in Experiment 2 is not new enough, and the proposed algorithm should be compared with the most typical algorithm.

3. The analysis of the experimental results is not sufficient in this manuscript, and it is necessary to explain why the improved algorithm have achieved good results.

4Explain why the meaning of "GHA - * -A /b" was introduced in line 275, and "GHA - * -s /B"? Is it used in the table?

5There are many clerical errors in the paper there. For example, "GT" opens line 102 with "GTN"; "Figure 2©" in line 216 should read "Figure 2 (c)"; "CIFAR10,100" should read "CIFAR10/100" in line 19.

Comments on the Quality of English Language

English should be improved.

Author Response

Thank you for your valuable comments. We analyzed the comments you pointed out and revised the text accordingly. The reflected corrections were responded to in 'response' one by one, and the modified parts were marked in red in the text. Thanks again for your efforts.

Reviewer 2 Report

Comments and Suggestions for Authors

This paper proposes a GHA (graph head attention) for replacing the computationally expensive MHA module in the VIT structure that ignores local relationships, and GHA-ViT outperforms pure ViT-based models on small-sized CIFAR10, 100, MNIST, MNIST-F datasets, and the medium ImageNet-1K dataset.

However, many areas require further clarification and improvement:

1. In the induction part, the first paragraph describes many improvement methods to solve the problems of VITs high training cost (time, quantity), neglect of the locality of an image, etc. Each of them has its innovations. You summarize the limitations of these methods: not properly considering the spatial geometric relationship between local regions or between global and local regions and having limitations in reducing the number of computations because they depend heavily on the combination of multi-head attention (MHA). However, is the correspondence between this and the methods you listed rigorous? For example, the description of gMLP and FNet cannot support the limitations you summarized.

2. This article has repeatedly emphasized that the VIT-based method ignores the relationship between the locality and globality, but there is no relevant text describing its importance. It is recommended to add relevant explanations.

3. Equation (1) illustrates the input data redefined by attention score, W represents the trainable parameter for the node feature ℎ, which is applied to the current node and between nodes at the same time. If the dimension and rules of W can be provided, it will be more helpful in understanding the formula.

4. The description of the attention matrix (AT) calculation process in Section 4.1 contains the sentence " Q, K, and V can be obtained through the dot product of input H and the weight matrices. " This H is associated with H_{out} in formula (14) if being able to clearly point out the connection between the two in the description can help readers understand the meaning of the article more easily.

5. Figure 3 shows the visualization of the two graph generation methods, which has some problems: first, the position display (yellow and red triangles) of the patch is relatively blurry, and the image ratio of the patch in the red and yellow box areas is not unified. Secondly, the strength of the relationship between patches in the visualization does not clearly correspond to the description of the differences between the two methods in Section 5.5, and there is a lack of representative and quantitative evaluation indicators. If Figure 3 can be modified and more representative examples can be found, it will help support the relevant advantages of GHA.

6. The author may include the following reference: Transformers in vision: A survey; Weight Asynchronous Update: Improves the Diversity of Filters in Deep Convolutional Network; Learning Motion Representation for Real-Time Spatio-Temporal Action Localization; A survey of visual transformers.

Based on the above points, I suggest revising and adding to the paper. The authors should address the above issues and provide more clarity and detail.

Author Response

(The authors gave the same response as above.)

Reviewer 3 Report

Comments and Suggestions for Authors

The paper proposes an improved verstion of the typically utilized ViT, which replaces the multihead attention with a few graph head attension. Experimental results have solidly verify the performance of the propose method on several datasets compared with a few typically used baselines. It provides insights for utilizing ViT based methods in real application. The paper is well represented and organized. Below are some suggestions to improve the paper for the journal publication:

(1) The abstract part can be improved by adding the summarization of the performance of GHA-Vit, which can help highlight the general experimental evalution.

(2)It's better to explain GHA-B in Abstract, giving the meaning of B.

(3) 3.1 Vistion Transformer, 3.2 Patch Generation, and 3.3 Graph Attention Networks are better to be involved in Section 2. Related Studies.

(4) In Table 1. Model GHA-*-a/b, the * (can be S and B) are not explained.

Author Response

(The authors gave the same response as above.)
